# Multi-step rainfall forecasting using deep learning approach

Sanam Narejo[1], Muhammad Moazzam Jawaid[1], Shahnawaz Talpur[1], Rizwan Baloch[1] and Eros Gian Alessandro Pasero[2]

[1] Department of Computer Systems, Mehran University of Engineering and Technology, Jamshoro, Sindh, Pakistan
[2] Department of Electronics and Telecommunication (DET), Politecnico Di Torino, Turin, Italy



## ABSTRACT

Rainfall prediction is immensely crucial in daily life routine as well as for water resource management, stochastic hydrology, rain run-off modeling and flood risk mitigation. Quantitative prediction of rainfall time series is extremely challenging as compared to other meteorological parameters due to its variability in local features that involves temporal and spatial scales. Consequently, this requires a highly complex system having an advance model to accurately capture the highly non linear processes occurring in the climate. The focus of this work is direct prediction of multistep forecasting, where a separate time series model for each forecasting horizon is considered and forecasts are computed using observed data samples. Forecasting in this method is performed by proposing a deep learning approach, i.e, Temporal Deep Belief Network (DBN). The best model is selected from several baseline models on the basis of performance analysis metrics. The results suggest that the temporal DBN model outperforms the conventional Convolutional Neural Network (CNN) specifically on rainfall time series forecasting. According to our experimentation, a modified DBN with hidden layes (300-200-100-10) performs best with 4.59E−05, 0.0068 and 0.94 values of MSE, RMSE and R value respectively on testing samples. However, we found that training DBN is more exhaustive and computationally intensive than other deep learning architectures. Findings of this research can be further utilized as basis for the advance forecasting of other weather parameters with same climate conditions.

## INTRODUCTION

Anticipating the future values of an observed time-series phenomena plays a significant role to improve the quality of services. For instance, accurate predictions can greatly revolutionize the performance in the arena of medical, engineering, meteorology, telecommunication, control systems, business intelligence, crypto-currency and most important the financial outcomes. Anticipating adequate predictors and indicators from historical data requires statistical and computational methods for correlating dependencies. Specifically, between past and future values of observed samples and techniques to cop up with longer horizons (*Bontempi, Taieb & Le Borgne, 2012*). Over the

Corresponding author
Sanam Narejo,
sanam.narejo@faculty.muet.edu.pk

last few decades, the research community has shown an increasing interest in the time series analysis, modelling, prediction and forecasting. However, future prediction remains challenging due to the complex nature of problem.

It is important to mention that weather forecasting is significant not only for individual's everyday life schedule, but also for agriculture sector as well as several industries. These forecasts can also help decision-making processes carried out by organizations for disaster prevention. Being specific, rainfall is significant for agriculture, food production plan, water resource management and likewise other natural systems (*Bushara & Abraham, 2015*). The variability of rainfall in space and time, however, renders quantitative forecasting of rainfall extremely difficult (*Luk, Ball & Sharma, 2001*). The behaviour and structure of rainfall including its distribution in temporal and spatial dimensions depends on several variables, for instance, humidity, pressure, temperature and possibly wind direction and its speed. Apart from this, a time series of rainfall usually contains local features too, for example, bursts of heavy rain between prolonged low intensity rainfall duration. In real, these local features are not fixed in a time slot which renders the prediction of occurrence more difficult.

Since decades, the atmospheric forecasting was calculated through physical simulations in which the present state of the atmosphere is sampled, and future state is computed by numerically solving the equations of fluid dynamics and thermodynamics. Usually, the temporal and spatial characteristics of rainfall forecast rely heavily on the hydrological requirements. A hydrological model is characterization of a real-world hydrologic features, for example, water surface, soil water, wetland, groundwater, estuary. This type of modelling supports in managing, predicting, evaluating and understanding water resources by developing small-scale computer simulations, constructing physical models and mathematical analogues. This indicates that hydrological process models can be divided into three major categories, specifically, physical, conceptual and metric-based or computational models (*Beck, Kleissen & Wheater, 1990*). Conceptual modelling is simply a graphical visualization and representation of real world model using hydrological components. It is used to determine what aspects of the real world to include, and exclude, from the model, and at which level of detail, the model will demonstrate. On the other hand, physical models focus more towards the underlying physics using mathematical equations for hydrological process. Despite their good performance, these hydrological models, conceptual and physical do not perform well when applied to periods with climate conditions that differ from those during model calibration (*Bai, Liu & Xie, 2021; Duethmann, Blöschl & Parajka, 2020; Xu et al., 2020*). Thus, one of the possible solutions might be to select site-specific model, which includes non-hydrostatic cloud physics. Nevertheless, the black-box models are an alternative and more successful approach for modeling complex and nonlinear hydrological processes.

Moreover, in order to forecast the rainfall using physical-based process, model becomes unfeasible due to the complexity of the atmospheric processes by which rainfall is generated. In addition, the lack of data on the necessary temporal and spatial scales affects the prediction process (*Cristiano, Ten Veldhuis & Van de Giesen, 2017*). Thus, all these factors make rainfall time series prediction more challenging task as compared to other

meteorological parameters. Accordingly, we believe that the requirements for such a highly complex system should include an advance model to accurately capture the highly non linear processes occurring in the climate.

The size of forecasting horizon is enormously critical and is considered as one of the important feature in prediction process. One-step forecasting of a time series is already a challenging task, performing multi-step, i.e, **h-steps** ahead forecasting is more difficult (*Längkvist, Karlsson & Loutfi, 2014*) because of additional complications, such as accumulation of errors, reduced accuracy, and increased uncertainty (*Kuremoto et al., 2014*). Generally, on broader spectrum multistep forecasting can be computed through two major strategies. The first is recursive approach and the second one is direct approach. In recursive approach, multi step forecasting is handled iteratively. This means a single time series model is developed and each subsequent forecast is estimated using previously computed forecasts. On the other hand, the direct approach establishes a separate time series model for each forecasting horizon and forecasts are estimated directly by implemented models. However, the choice of selection in between of these two strategies involves a trade-off between bias and variance (*Taieb & Hyndman, 2014*). Conventionally, multistep forecasting has been managed recursively, where a model is setup as one step forecasting model and each forecast is estimated using previous forecasts.

Nevertheless, one cannot ignore the fact that minimization of 1-step forecast errors is not guaranteed to provide the minimum over textbfh-steps ahead errors. In this current research, the emphasis is on direct prediction of multistep forecasting, where a separate time series model for each forecasting horizon is considered and forecasts are computed using the observed data samples. In fact, the direct strategy minimizes the **h-step** ahead errors instead of considering one-step ahead. Huge number of studies comparing recursive and direct forecasting strategies are present in literature; for further details, see (*Tiao & Tsay, 1994*; *Cheng et al., 2006*; *Hamzaçebi, Akay & Kutay, 2009*; *Kline, 2004*; *Kock & Teräsvirta, 2011*). It is also apparent from the literature that the simple time series models contain no hidden variables.

In general terms, the fully observed models depend upon two types of variables: the first one is vector autoregressive and the subsequent one is Nth order Markov model. Despite of the simplicity, these models are constrained by their lack of memory (*Taylor, Hinton & Roweis, 2011*). Initially, classic time series modeling was performed by using autoregressive integrated moving average (ARIMA) and seasonal ARIMA (SARIMA) (*Zhang, Patuwo & Hu, 1998*; *Tseng, Yu & Tzeng, 2002*). However, these models are basically linear models (*Zhang, 2003*) and have a limited ability to capture highly nonlinear characteristics of rainfall series. Recent developments in artificial intelligence and, in particular, those techniques aimed at pattern recognition, however, provide an alternative approach for developing of a rainfall forecasting and run-off models (*Wu, Chau & Fan, 2010*; *Dounia, Dairi & Djebbar, 2014*; *Nourani, Tajbakhsh & Molajou, 2019*; *Nourani et al., 2019a*; *Ali et al., 2018*; *Ali et al., 2020b*; *Ali et al., 2018*). Artificial neural networks (ANNs), which perform a nonlinear mapping between inputs and outputs, are one such technique.

In particular, for rain prediction researchers in *Kashiwao et al. (2017)* predicted local rainfall in regions of Japan using data from the Japan Meteorological Agency (JMA). A multi-layer perceptron (MLP) is implemented with a hybrid algorithm composed of back-propagation (BP) and random optimization (RO) methods, and radial basis function network (RBFN) with a least squares method (LSM), and compared the prediction performance of the two models. Similarly, ANN shows superior result in comparison to the traditional modeling approaches in *Hung et al. (2009)*. In their research, results show that ANN forecasts achieved satisfactory results and have superiority over the ones obtained by the persistent model. Emotional artificial neural network (EANN) models have recently been developed and deployed by integrating artificial emotions and the ANN technique as a new generation of traditional ANN-based models. *Nourani et al. (2019b)*, proposed the first ever application of these models for multistep precipitation forecasting. Simultaneously, researchers have also suggested the long-term forecasting of precipitation using threshold-based hybrid data mining approach (*Nourani, Sattari & Molajou, 2017*) and a novel data-intelligent approach (*Ali et al., 2020a*).

It is important to mention that a multilayer ANN usually contains three layers: an input layer, an output layer, and one or more hidden layer. The hidden layer is useful for performing intermediary computations before mapping the input to the output layer (*Darji, Dabhi & Prajapati, 2015*). Prior to deep learning, problems involving more than two hidden layers were uncommon. With simple data sets, two or fewer layers are often adequate. ALbeit, additional layers may be useful in complex datasets involving time series or computer vision applications. Artificial neural networks with many hidden layers forms a deep architecture composed of multiple levels of non linear operations. Training those deep architectures comes under the umbrella of Deep Learning. When a neural network is composed of more than one or two hidden layers, contingent upon that situation the higher layers compose abstractions on the top of previous layers. Deep Learning Architectures are able to extract high level abstractions from input distribution of data by means of multiple processing layers, composed of multiple linear and non-linear transformations.

To summarize, a number of forecasting approaches have been reported in literature as recent advancements in computing technologies combined with the growing availability of weather-related data has served to dramatically improve the accuracy of forecasts. Recent literature demonstrates that deep learning models are excelling on the platform of machine learning algorithms for time series prediction (*Hinton, Osindero & Teh, 2006*; *Bengio et al., 2007*; *Mohamed et al., 2011*; *Mohamed, Dahl & Hinton, 2011*; *Seide et al., 2011*; *Bordes et al., 2012*; *Glorot, Bordes & Bengio, 2011*; *Hernández et al., 2016*; *Busseti, Osband & Wong, 2012*; *Liu et al., 2014*; *Dalto, Matuško & Vašak, 2015*; *He, 2017*; *Kuremoto et al., 2014*; *Längkvist, Karlsson & Loutfi, 2014*; *Narejo & Pasero, 2017*); however, an accurate forecasting of rainfall is still challenging in context of hydrological research (*Hong, 2008*). Accordingly, an attempt is made in this work for multi-step rainfall prediction using deep learning approach.

**Table 1 Deep learning architectures and their applications domain.**

| Deep learning architectures | Excelled domains |
|---|---|
| CNN | Speech recognition, image recognition and classification, human pose and 3D human action recognition |
| DBN | Dimensionality reduction, image recognition, information retrieval, natural language understanding, prediction problems, feature extraction |
| SAE | Data compression, dimensionality reduction and representation learning, document clustering, sentiment analysis, image processing and object recognition |
| RNN\LSTM | Sequence prediction, online handwriting recognition, gesture recognition, natural language and speech processing, human activity recognition, automatic translation of texts and images |

## Main contribution of this research

DBNs are effective models for capturing complex representations mostly from static and stationary data such as image classification and object recognition. DBNs actually lacks the dynamic modelling and are not accurately adequate for non-stationary environments based on time variant features. In *Narejo & Pasero (2016)*, researchers have proposed a hybrid approach for time series forecasting of temperature data using DBN and Nonlinear Autoregressive Neural Networks (NARX). The authors employed DBN for feature extraction whereas NARX network was developed and trained for extrapolating the temporal forecasts. On the contrary, in the current research, we propose a simple extension to DBN-RBM model in order to capture temporal dependencies for multi-step ahead rainfall prediction. Additionally, the extended model is now capable to forecast multi-steps ahead, rather than just performing prediction for next one step ahead. The extended model still maintains its most important computational properties, such that exact inference and efficient approximate learning using contrastive divergence. Comparative analysis is also conducted by comparing the performance metrics with other state of the art deep learning models.

## RESEARCH BACKGROUND

The key focus of deep learning is, to automatically discover the hierarchical representations of data, from lowest level features to high level concepts. This automatic learning of features at multiple levels of abstraction influences to learn complex functions mapping the input to the output directly from data, independent of human-crafted features.. Deep learning is not a single approach, rather it is a class of algorithms and topologies including Recurrent Neural Networks (RNNs), Convolutional Neural networks (CNNs), Deep Belief Networks (DBNs), Long short term memory (LSTMs), and stacked Auto-Encoders (SAEs). These approaches are applied to solve a a broad spectrum of problems. Although, depending upon their architecture, training and nature of the problem, these models achieve breakthrough performance in specific domains as summarized in Table 1. Apart from this, a number of complex problems can be solved by employing these models in combination. For instance, human activity recognition, document analysis, processing and labelling video frames. In this context, we present theoretical concepts of selected deep learning models and their implementation over

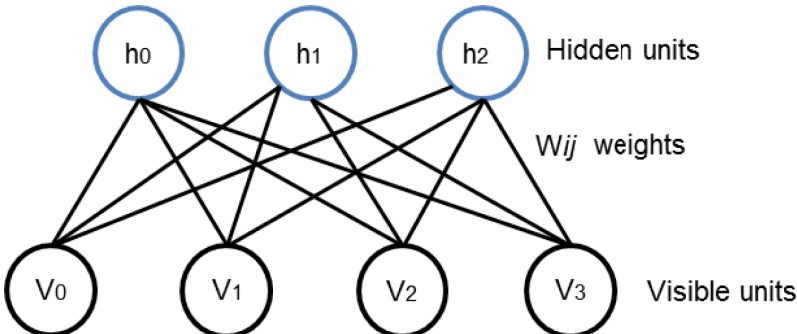

**Figure 1 Two layered RBM with hidden and visible units.** The visible units are responsible to take the input data. the hidden units working on generating observations of model dependencies. Subsequently, the process is back and forth for understanding the patterns and underlying structure of provided data.

time-series data. Moreover, we present our implementation for training these models over time series meteorological data.

From structural point of view, deep learning is in fact adding more hidden layers to the neural network architecture. The depth of architecture denotes the depth of the graph, i.e, the longest path or number of levels from input node to output node. Practical implementations indicate that training these architectures is much more challenging and difficult than the shallow architectures (*Bengio et al., 2007*; *Erhan et al., 2009*). The gradient based training of deep supervised multi-layer neural networks with random initialization of network parameters often gets stuck in local minima. Consequently, it becomes difficult to obtain good generalization. Therefore, the training strategy for deep neural network has been modified in this work to distinguish from the reported studies. In this context, a number of deep architectures are discussed below.

## Deep belief network

DBNs are intended to be one of the foremost non-Convolutional models to successfully admit the training of deep architectures. DBN has played Key role in the revival of deep neural networks. Earlier than the preface of DBN, deep models were hard to optimize (*Bengio, 2009*). The layered structure in DBN can be formed by stacking RBMs which are used to initialize the network in the region of parameter space that finds good minima of the supervised objective. RBM relies on two layer structure comprising on visible and hidden nodes as shown in Fig. 1. The visible units constitute the first layer and correspond to the components of an observation whereas the hidden units model dependencies between the components of observations. Then the binary states of the hidden units are all computed in parallel using (1). Once binary states are chosen for the hidden units, a "reconstruction" is achieved by setting each vj to 1 with a probability given in (2).

$$p(h_i = 1|v) = sigmoid\left(\sum_{j=1}^{m} w_{ij}v_j + c_i\right) \tag{1}$$

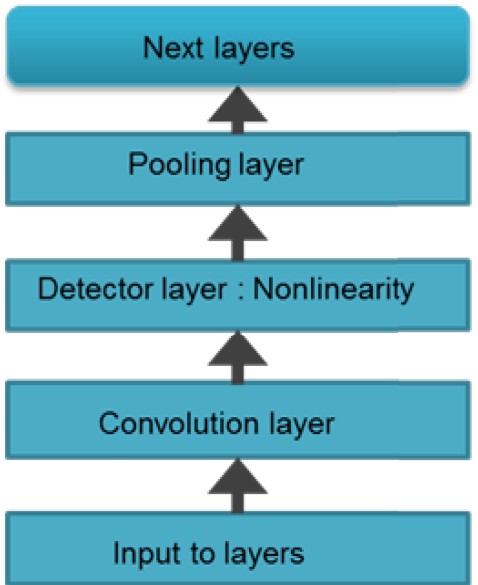

**Figure 2 Layered architecture of CNN.** A generalized architecture of CNN model is presented with conventional layers. Apart from the input layer and the next layer, most noteable ones are convolutional layer, detector layer and a pooling or subsampling layer.

$$p(v_j = 1|h) = sigmoid\left(\sum_{i=1}^{n} w_{ij}h_i + b_j\right) \qquad (2)$$

The weight wij can be updated using difference between two measured data dependent and model dependent expectations as expressed in Eq. (3). Where $\varepsilon$ is a learning rate.

$$\Delta wij = \varepsilon(<vjhi>_{data} - <vjhi>_{recon}) \qquad (3)$$

The DBN model is trained by training RBM layers using contrastive divergence or stochastic maximum likelihood. The parameters of RBM then designate the parameters of first layer of the DBN. The second RBM is trained to model the distribution defined by sampling the hidden units of the first RBM whose visible layer is also working as an input layer as well. This procedure can be repeated as desired, to add as many layers to DBN.

## Convolutional neural network

CNNs are made up of neurons that have learnable weights and biases. Each neuron receives some inputs, performs a dot product and optionally follows it with a non-linearity. The whole network still expresses a single differentiable score function. Convolution is a mathematical concept used heavily in digital signal processing when dealing with signals that take the form of a time series. To understand, CNN is a deep network where instead of having stacks of matrix multiply layers, we are going to have stacks of convolutions. As it can be seen in Fig. 2, three main types of layers are used to build ConvNet architectures: Convolutional Layer, Pooling Layer, and Fully-Connected Layer or Next layers. Convolution is basically combined integration of two functions as equated in (4)

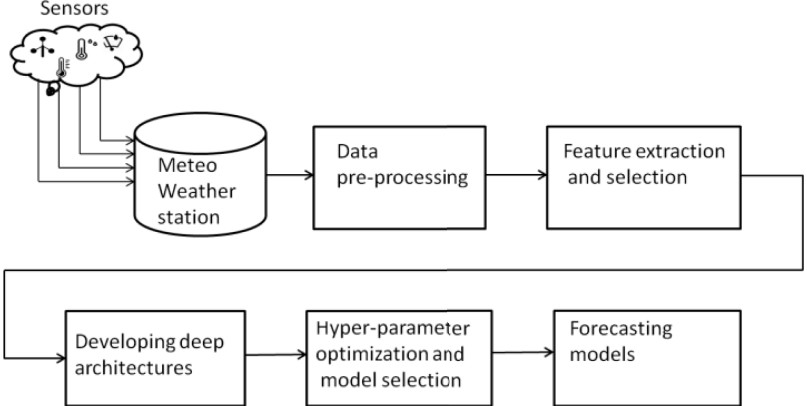

**Figure 3 Research Methodology.** Initially the data recorded by sensor through Meteo weather station is downloaded via Neuronica Lab resources.The next step is to apply pre-processing and perform feature extraction accordingly. Finally, training and optimizing the deep learning model as forecasting model.

$$S(t) = \int (a)w(t-a)da \tag{4}$$

The convolution operation is typically denoted by asterisk as shown in (5)

$$S(t) = (x * w)(t) \tag{5}$$

The first arguments x to the convolution is often referred to as input and the second argument w as the kernel. The output s(t) is sometimes referred to feature map. In ML applications, the input is usually a multidimensional array of data and the kernel is usually a multidimensional array of parameters that are adapted by learning algorithm. The Next layers can be the same conv-nonlinear-pool or can be fully connected layers before output layer.

To summarize, convolutional mechanism to combine or blend two functions of time in a coherent manner. Thus, the CNN learns the features from the input data. Consequently, the real values of the kernel matrix change with each iteration over the training, indicating that network is learning to identify which regions are of significance for extracting features from the data.

## RESEARCH METHODOLOGY

Figure 3 presents the general overview of methodology employed in this work. The real time meteorological data was downloaded from Meteo weather station installed at Neuronica Laboratory, Politecnico Di Torino (*Narejo & Pasero, 2017*) as shown in Fig. 4. Recorded data contains several meteorological parameters such as however our primary concern for the current research was rainfall forecasting. In order to compute the accurate forecast, the foremost step was data analysis. This analysis was performed by applying some pre-processing steps over the experimental data i.e data filteration, noise removal and outlier detection as follows.

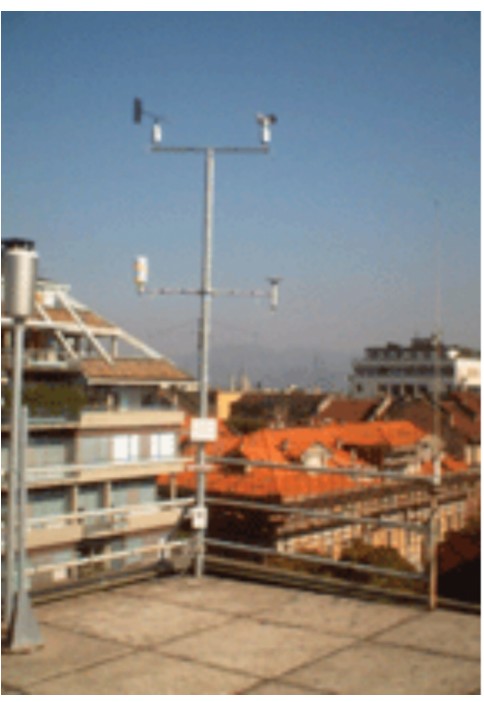

**Figure 4 Meteo Weather station at Politecnic Di Torino.** The time series data is recorded from the Meteo weather station mounted at the top of DET building and connected with Neuronica Laboratory, Politecnico Di Torino. (*Narejo & Pasero, 2017*).

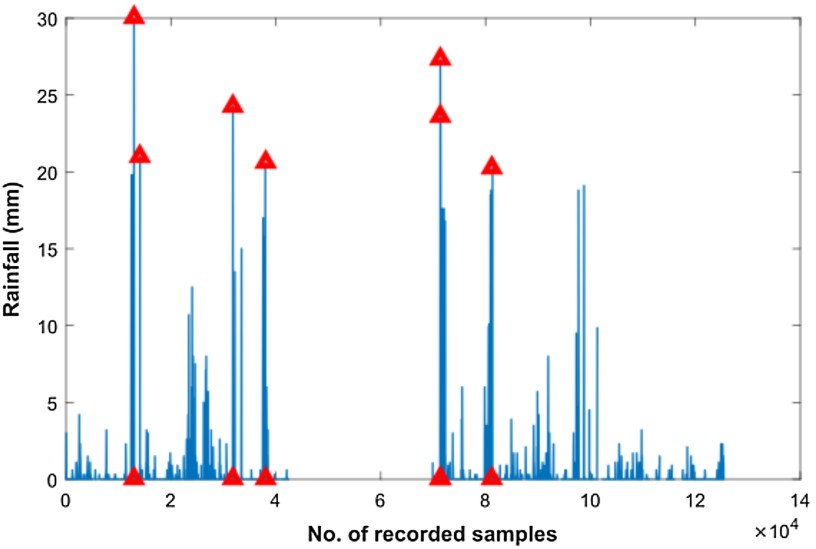

**Figure 5 Outlier detection in Rainfall time series data.** Outliers are highlighted with red triangles in Rainfall time series data.

## Filtering and noise removal

In order to smoothen the time series rain data and to normalize noisy fluctuations, a number of different filters were applied. However, the first step towards filter applications was outlier detection in our rain dataset as shown in Fig. 5. Subsequently, we filtered the

**Table 2  Mean square error on filtered rain data.**

| Filter | MSE |
| --- | --- |
| Median | 0.0569 |
| Moving Average | 0.0447 |
| Low-pass Butterworth | 0.0352 |
| Savitzky golay | 0.0302 |

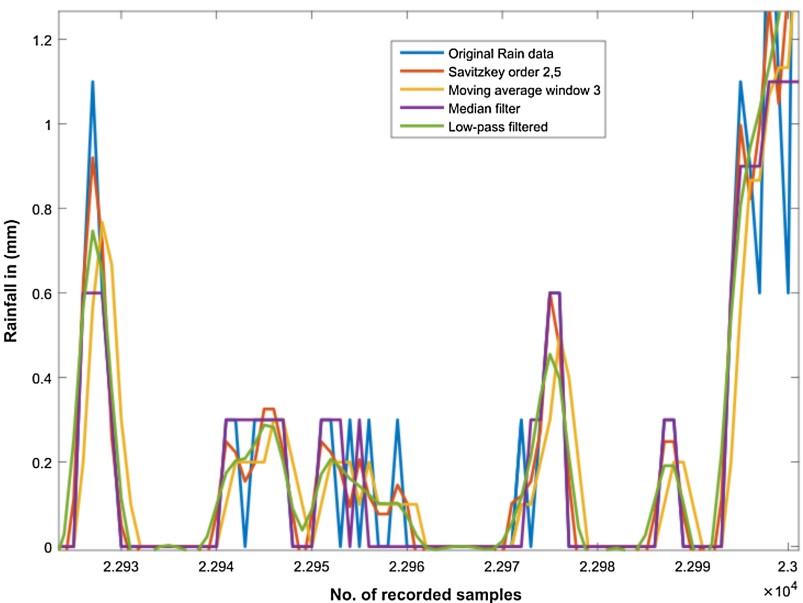

**Figure 6  Filtering the rainfall time series.** It can be observed from figure that original rain data is highly fluctuated and sharp edges. This is further smoothened and sharp edges are reduced by applying different filters as shown in the figure.     

rain data using Moving Average, Savitzkey golay and other low pas filters as presented in Table 2. The original and filtered rain data is presented in Fig. 6 to demonstrate the effectiveness of the pre-processing step. However, when we trained our temporal DBN models with these filtered data, it was observed that learning rate was much better for models based upon Moving Average and low pass filtered data. However, it was observed later that the input data with moving average filter introduces some delay in estimated rainfall predictions. Hence, we opted for lowpass filteration for subsequent experiments .

## Feature extraction

In order to compute accurate rainfall predictions, we must have some meaningful attributes that provides content contribution and possibly reduced error rates. Both the internal and external characteristics of rainfall field depend on a number of factors including pressure temperature, humidity, meteorological characteristics of catchments (*Nasseri, Asghari & Abedini, 2008*). However, the rainfall is one of the most difficult variables in the hydrologic cycle. The formation mechanism and forecast of rainfall involve

a rather complex physics that has not been completely understood so far (*Hong & Pai, 2007*). In order to resolve this, we put some more efforts while creating significantly relevant feature set particularly for rainfall nowcasting. Accordingly, we investigated a number of different feature sets by adding and deleting the meteorological parameters in sequence. Subsequently, finding the appropriate lagged terms of selected parameters to be used included as features.

Successively, we also calculated some statistical features considering mean, standard deviation, variance, maximum, minimum, skewness and kurtosis. We tested our feature sets by training some models and later on, we found that the DBN models were performing comparatively better if we exclude skewness and kurtosis from the selected features. Hence, the finalized features to predict rainfall at (t+h) were:

$$
\begin{aligned}
Rain(t+h) = [&rain(t), rain(t-1), rain(t-2), rain(t-3), mean(t:t-3), \\
&std(t:t-3), humidity(t), pressure(t), temperature(t), humidity(t-1), \\
&pressure(t-1), temperature(t-1), humidity(t-2), \\
&pressure(t-2), temperature(t-2)]
\end{aligned}
\tag{6}
$$

where, **h** in Eq. (6) is a selected horizon for forecasting. In our case, it was fixed as 1, 4, and 8 indicating for the next sample, for the next 1 h and the next 2 h respectively in future. It is highly important to reiterate that the frequency of our time series recorded data is 15 min. Sensor generates the value after every 15 min. Therefore, in next 1 h, 4 samples being recorded. With in 2 h, 8 samples. Also, the subtraction in the equation, let it be considered as "**-n**" is indicating that previous n samples in the series. let us suppose that if $n = 1$, this suggests the one sample earlier than the current one in the series or the immediate previous sample in series. If $n = 2$, this suggests 2 previous samples next to the current sample will be chosen. Similarly, if $n = 3$, the three immediate previous samples from the series will be selected. Finally, in summary, in order to forecast rainfall for h steps ahead, the required input attributes as presented in (6) are the thre previous samples and one current value at time "t" of the rain data. Moreover, the mean and standard deviation of earlier mentioned four rain samples.Humidity, pressure and temperature at current time t also two previous samples of these variable as time t-1 and t-2.

## Experimental setup

The RBM models imitate static category of data and it does not integrate any temporal information by default. In order to model time series data, we added the autoregressive information as input by considering the previous lag terms in series. Here, in the proposed method, apart from autoregressive terms, we also incorporate some statistical dependencies from the temporal structure of time series data in the form of input feature set parameters. This was done due to the fact that, multistep forecasting for longer horizon is more challenging task. Therefore, some extra statistical considerations and computations needed to be done in order to understand the proper underlying behaviour of temporal sequence. This additional tapped delay of previous samples as input attributes, is shown in Eq. (6). It actually introduces temporal dependency in the model and further transforms the model from static to dynamical form. In general, the dynamic ANN

depends on a set of input predictor data. Consequently, the dataset needs to define and represent relevant attributes, to be of good quality and to span comparable period of time with data series (*Abbot & Marohasy, 2012*). The rainfall time series dataset is in total composed of 125691 no. of sample recordings. The dataset is divided into three parts prior to training. We divided 70% of the total data for training of the models, 20% for testing and rest was used to validate the h-step ahead forecasting.

## Selecting deep layered architecture

In machine learning while fitting a model to data, a number of model parameters are needed to be learned from data, which is performed through model training. Moreover, there is another kind of parameters that cannot be directly learned from the legitimate training procedures. They are called hyper parameters. Hyper-parameters are usually selected before the actual training process begins. The hyper-parameters can be fixed by hand or tuned by an algorithm. It is better to adopt its value based on out of sample data, for example, cross-validation error, online error or out of sample data. The classical recommendation of selecting two layered architecture for neural networks has been modified in this work with the advent of deep learning.

Deeper layers, or layers with more than two hidden layers, may learn more complex  representations (equivalent to automatic feature engineering). The number of neurons in the hidden layers is an important factor in deciding the overall architecture of the neural network. Despite the fact that these layers have no direct interaction with the outside world, they have a tremendous effect on the final outcome. The number of hidden layers as well as the number of neurons in each hidden layer must be considered carefully. Use of few neurons in the hidden layers will result in underfitting i.e. failure to adequately detect the signals in a complicated data set. On the other hand, using too many neurons in the hidden layers can result in several problems. First, a large number of neurons in the hidden layers may result in overfitting. Overfitting occurs when the neural network has extraordinary information processing capacity that the limited amount of information contained in the training set is not sufficient to train all of the neurons in the hidden layers. Even when the training data is adequate, a second issue may arise. An inordinately large number of neurons in the hidden layers can increase the time it takes to train the network. The amount of training time can increase to the point that it is impossible to adequately train the neural network. Obviously, some compromise must be reached between too many and too few neurons in the hidden layers.

The researchers have advocated in *Erhan et al. (2010)* that the reason for setting a large enough hidden layer size is due to the early stopping criteria and possibly other regularizers, for instance, weight decay, sparsity. Apart from this, the greedy layer wise unsupervised pretrainig also acts as data dependent regularizer. In a comparative study (*Larochelle et al., 2009*), authors found that using same size for all layers worked generally better or the same as using a decreasing size (pyramid like) or increasing size (upside down pyramid). They further argued that certainly this must be data dependent. However, in our research task the decreasing size structure worked far better than the other too. Consequently, this architectural topology was chosen as standard for further forecasting

models. The authors in *Narejo & Pasero (2018)* have argued that in most of the conducted experiments it was found that an over-complete first hidden layer in which the dimensions are higher than the input layer dimensions, works better than the under-complete one.

Due to the availability of high performance computing facilities and massive computational resources, the more productive and automated optimization of hyper parameter is possible through grid search or random search methods. We have applied both of the mentioned strategies in our experiments as discussed in the next section.

## RESULTS

For each forecasting horizon, a separate model is trained and optimized. However, as explained in earlier in Introduction section, multi-step forecasting is much more challenging than one-step ahead. Because, as the forecasting horizon is increased, it is obvious that the propagation of error in each sample will be raised. Due to this known fact, the performance accuracy for longer horizon is slightly less than that of the short forecasting horizon. While training and selecting the final model for each separate forecasting horizon, multiple models were developed and one for each forecasting horizon was finalized on the basis of performance evaluation in terms of RMSE, MSE and R parameters.

### One step ahead forecasting

The commendable deep RBM model for one step ahead rainfall forecasting was chosen with the architecture of (800-400-100-10) hidden layers resulting the depth of four levels. Apart from this, one input layer consisting of fifteen units and one output layer with one unit for predicting the target. The model performed well with RMSE of 0.0021 on training data and 9.558E−04 on test data set. The actual and the forecasted time series is plotted in Fig. 7.

### Four steps ahead forecasting

In order to perform four steps ahead forecasting the model with the following hidden layer dimensions (600-400-100-10) is proposed. Similar to earlier mentioned model, the input layer is created with 15 nodes and an output layer is connected for predictions. It resulted with RMSE of 0.0093 on training and 0.0057 on test data set. The actual and the forecasted rainfall time series can be seen in the Fig. 8.

### Eight steps ahead forecasting

Eight step ahead forecasting was more troublesome task than the rest two mentioned above. Considerably, more networks were attempted to be trained and to be selected as the optimal one. However, the accuracy of each model varies with very slight difference. The outcomes observed for each trained model were almost equivalent. A variety of architectures were optimized and performance measures are summarized in Table 3. It can be observed that the deep architecture of DBN-RBM with (300-200-100-10) as hidden layer dimensions shown in Fig. 9 was found to be the most appropriate and adequate for eight steps ahead forecasting.

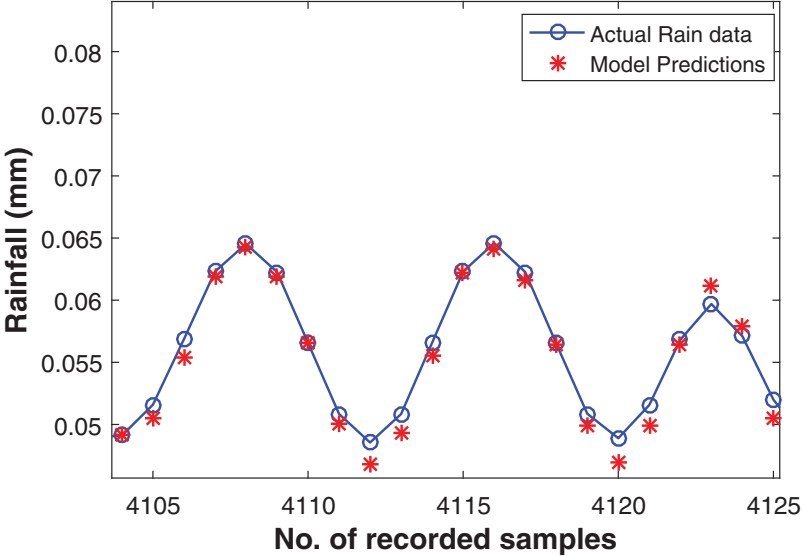

**Figure 7 Rainfall forecasting for next sample.** Actual rain samples and model predictions for one-step-ahead forecasting. The blue line with circle is representing the actual samples. The data in red is highlighting the estimations computed by model as one step ahead forecasting.

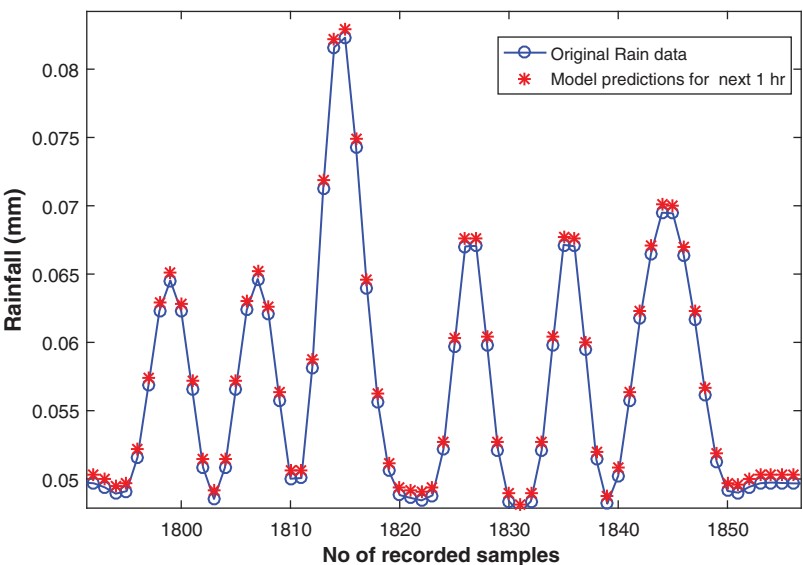

**Figure 8 Rainfall forecasting for next 1 h.** Actual rain samples and model predictions for one-step-ahead forecasting. The blue line with circle is representing the actual samples. The data in red is highlighting the estimations computed by model as one step ahead.

Figure 10 presents the forecasting of eight steps ahead rainfall time series. It can be observed in the figure that the forecasting samples are not exactly replicating the original data indicated by blue circles. Apart from this there is some sort of delay. This delay is due to the prediction of longer multi-step samples. Forecasting for longer horizon is an arduous task, therefore a deep CNN model is also introduced for forecasting in this section

Table 3 Performance measures MSE, RMSE and R of proposed deep architectures for eight step ahead rainfall forecasting.

| S.No | Hidden layers | RMSE | | MSE | | R |
|------|---------------|----------|--------|------------|------------|-------|
| | | Training | Test | Training | Test | |
| 1 | 600-400-100-10 | 0.0187 | 0.0068 | 3.4842E−04 | 4.6692E−05 | 0.943 |
| 2 | 500-500-100-10 | 0.0188 | 0.0070 | 3.5372E−04 | 4.9085E−05 | 0.941 |
| 3 | 300-200-100-10 | 0.0182 | 0.0068 | 3.3915E−04 | 4.5971E−05 | 0.944 |
| 4 | 800-400-100-10 | 0.0184 | 0.0070 | 3.4021E−04 | 4.9127E−05 | 0.944 |
| 5 | CNN | 0.0209 | 0.0072 | 4.3475E−04 | 5.1615E−05 | 0.929 |

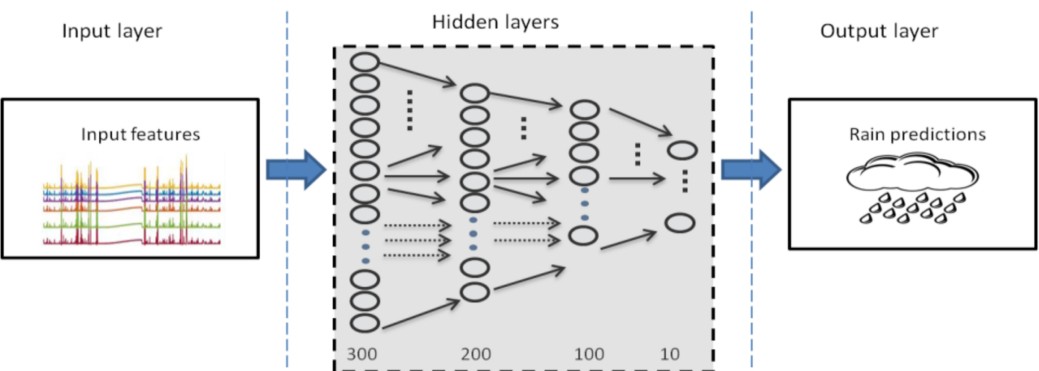

Figure 9 Optimal DBN model for rainfall prediction for eight steps ahead forecasting.

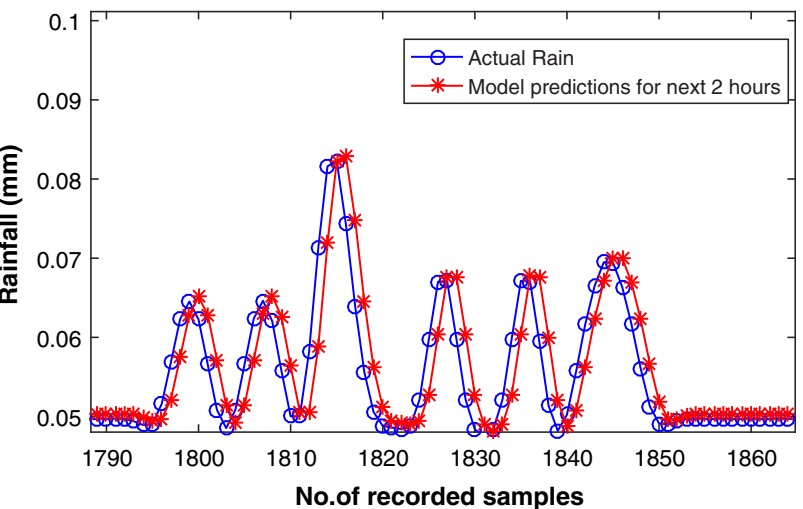

Figure 10 Rainfall forecasting for next 2 h. Actual rain samples and model predictions for next 2 h, i.e, eight steps ahead forecasting according to the frequency of our time series dataset. The blue line with circle is representing the actual samples. The data in red line is highlighting the estimations computed by model as eight-step-ahead forecasting.

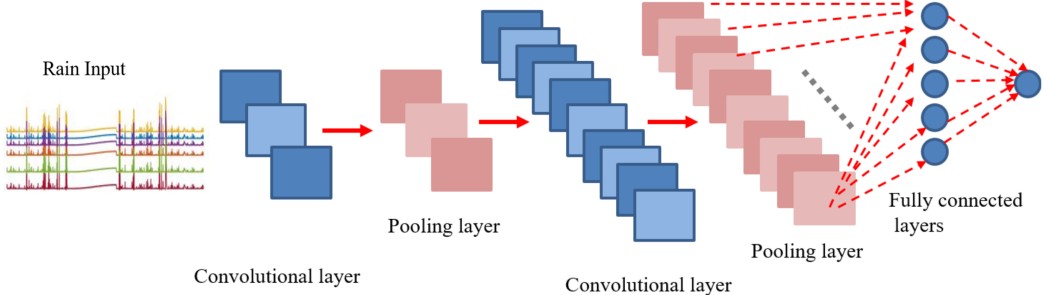

**Figure 11 CNN architecture for eight steps ahead rain forecasting.**

of our research activity. The latest literature exhibits that the structurally diverse CNN stands out for their pervasive implementation and have led to impressive results (*Cui, Chen & Chen, 2016*; *Krizhevsky, Sutskever & Hinton, 2012*; *Schroff, Kalenichenko & Philbin, 2015*).

In CNN model, the convolution filter or kernel is basically an integral component of the layered architecture. The kernels are then convolved with the input volume to obtain so-called activation maps. Activation maps indicate activated regions, i.e. regions where features specific to the kernel have been detected in the input data. In general, the kernel used for the discrete convolution is small, this means that the network is sparsely connected. This further reduces the runtime inference and back propagation in the network. CNN also typically include some kind of spatial pooling in their activation functions. This helps to take summary statistics over small spatial regions of input in order to make the final output invariant to small spatial translations of the input. CNNs have been very successful for commercial image processing applications since early.

In contrast to image classification, the modified version of conventional CNN is applied to time series prediction task for eight steps ahead forecasting of rainfall series. The proposed CNN includes four layers as shown in Fig. 11. The first convolutional layer was developed by thre filters with kernel size of (3, 1). Similarly, the second conv layer contained 10 filters with the same size as earlier (3, 1). The pooling layer was added by following the "average" approach for sub-sampling. However, in our case the averaging factor was unity. For fully connected layers, tangent hyperbolic activations were used followed by a linear layer for output predictions. To find out the accurate forecasting model, it is far important to evaluate and compare the performance of trained models. The natural measure of performance for the forecasting problem is the prediction error. MSE defined in Eq. (7) is the most popular measure used for the performance prediction (*Zhang, 2003*; *Ribeiro & Dos Santos Coelho, 2020*; *Ma, Antoniou & Toledo, 2020*; *Aliev et al., 2018*). However, the use of only one error metric (MSE) to evaluate the model performance actually lacks to represents the entire behaviour of the predictions in a clear way. Therefore, more performance measuring criteria should be considered to validate the results Hence, performance for each predictive model is quantified using two additional performance metrics, i.e. Root Mean Squared Error (RMSE) and Regression parameter R on Training and Test sets.

$$MSE = \sum_{i=1}^{n} \left( \frac{E_t}{N} \right)^2 \tag{7}$$

Where, N is the total number of data for the prediction and Et is the difference or error between actual and predicted values of object t. Table 3 presents the details related with the performance of each model for eight steps ahead forecasting. It can be observed from the table that the third model with the deep architecture of (300-200-100-10) layers, stands optimal in terms of all three performance metrics. The R parameter is linear regression, which relates targets to outputs estimated by network. If this number is near to 1, then there is good correlation between targets and outputs which shows that outputs are approximately similar to targets .

## DISCUSSION

In weather forecasting, specifically rainfall prediction is one of the most imperatives, demanding, critical operational problem. It is complex and difficult because, in the field of meteorology decisions are taken with a degree of uncertainty. This actually happens due to chaotic nature of the atmosphere which limits the validity of deterministic forecasts. Generally, the required parameters to predict rainfall are extremely complicated and highly variable. This increases the uncertainty in rainfall prediction task even for the shorter horizons (*Bushara & Abraham, 2013*). It is important to mention that it needs much more effort to compare and contrast different types of existing rainfall forecasting models as reported methods usually provides the comparison of their output with observed values. Thus, this evaluation becomes data-dependent due to the difference of data taken for the different regions and time periods.

In this context, we also trained some significant nonlinear autoregressive neural networks on our data. As our research work is based on time series forecasting of rainfall, the forecasting is done for three different forecasting horizons, next immediate value, the value of rain variable after 1 h, the value of rain variable for next 2 h. To develope and to further train the models efficiently, we selected seperate model for each forecasting horizon. This was done due to the data dependency available in the historical samples and also to produce accurate forecasting correspondingly. Despite of training deep learning architectures which automatically extract the meaningful features and patterns, we applied sufficient efforts to compute some statistical features for each forecasting horizon separately prior to giving the input data to the deep learning models. Number of attempts were taken to produce the deep learning model as accurate forecaster based on different architecture and different parameter settings. In partcular, we have only mentioned some optimal models in the result sections. The performance of each model was computed using MSE, RMSE and R.

## CONCLUSIONS

The paper presents rainfall time series forecasting of a specific site in Italy using deep learning architectures.. Deep learning architectures are accelerating rapidly in almost every

field of interest and replacing several other machine learning algorithms. Consequently, this gave us direction to further investigate these deep architectures over time series rainfall forecasting. Therefore, the primary focus of this research was to perform multi-step forecasting for rainfall data as its much more challenging than single-step ahead. During our research, it was observed that the parameters required to predict rainfall were enormously complex and subtle even for a short term period. Thus, different combinations of inputs and statistical features were investigated. The results presented in Table 3 indicate that DBN outperforms the conventional CNN model when larger forecasting horizon was considered.

It is important to mention that error measures play an important role in calibrating or refining a model in order to forecast accurately for a set of time series. Thus, three different performance metrics were considered for comparative analysis over the trained models. Considering the obtained RMSE and MSE values of trained models, it is obvious that deep learning architectures are significantly improving the test errors in contrast with the training errors.

During the training phase of models, it was observed that the Deeper architectures are more exhaustive as far as the computational resources are concerned. Due to this, it took almost more than couple of weeks to well train the deep hierarchical models on High Performance Computing (HPC). However, our major concern was not about acceleration but accurate modelling of data. Albeit, from future perspective the acceleration can be improved by utilizing the GPUs and FPGAs for similar implementations. In context of the future works, we believe that findings of this research can be further utilized as basis for the advance forecasting of weather parameters with same climate conditions.

## ACKNOWLEDGEMENTS

The authors are thankful to Prof. Eros Pasero , Neuronica laboratory of Politecnico Di Torino (Italy) for providing data and computational resources. Partial Computational resources were also provided by HPC @ POLITO and Mehran University of Engineering and Technology.

### Funding

The authors received no funding for this work.

### Competing Interests

The authors declare that they have no competing interests.

### Author Contributions

- Sanam Narejo performed the experiments, authored or reviewed drafts of the paper, and approved the final draft.

- Muhammad Moazzam Jawaid conceived and designed the experiments, performed the experiments, performed the computation work, authored or reviewed drafts of the paper, and approved the final draft.
- Shahnawaz Talpur performed the experiments, analyzed the data, authored or reviewed drafts of the paper, and approved the final draft.
- Rizwan Baloch analyzed the data, performed the computation work, prepared figures and/or tables, and approved the final draft.
- Eros Gian Alessandro Pasero conceived and designed the experiments, authored or reviewed drafts of the paper, and approved the final draft.

## Data Availability

Raw data and MATLAB code are available as Supplemental Files.

## Supplemental Information

Supplemental information for this article can be found online at http://dx.doi.org/10.7717/peerj-cs.514#supplemental-information.

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
