# Peer review of "Multi-step rainfall forecasting using deep learning approach"

_PeerJ Computer Science, doi:10.7717/peerj-cs.514_

## Round 0.1 · original submission · Major Revisions

After careful consideration of reviewers' comments, I recommend a major revision of the paper. Authors need to provide more information to enhance the delivery of the methods and perform some additional experiments. Please note that acceptance will be contingent on addressing sufficiently the concerns of the reviewers.

Reviewer 1 ·

Basic reporting

no comment

Experimental design

no comment

Validity of the findings

no comment

Additional comments

The manuscript presents "Multi-step Rainfall Forecasting using Deep Learning Approach", which is interesting. The paper has the potential to be published in this journal after revision. Appropriate revisions to the following points should be undertaken in order to justify recommendation for publication

1- The language of the manuscript should be checked and more proofreading is necessary. I also recommend that authors should refrain from using long sentences. Instead, they should use clear and shorter sentences in terms of the readability of the manuscript. In addition, please correct the grammatical errors throughout the manuscript.
2- Introduction chapter requires the research purposes in detail. Since they are vague in their present form, please revise them. The authors need to provide a very detailed statement as to how this paper differs from their other papers of similar titles.
3- Full names should be shown for all abbreviations in their first occurrence in texts. For example, CNN, DBN etc. (Noy only in Abstract).
4- Authors should give some comparisons about the usage of data-driven models a little bit more (briefly) in the "Introduction" section. They should mention about the performance of these models, shortly. It is well known fact that most of the statistical models (ANN, ARIMA, SVR, etc.) can predict one day ahead forecast with considerable accuracy. However, their performance reduces significantly when forecasting is made multiple days ahead. In this study the authors considered multi-step ahead forecast. It is important to see the performance of the model when no observed rainfall data were used for the entire validation period to forecast the rainfall data.
5- There are several conceptual/physically based models available for rainfall forecasting, which needs to be discussed in the introduction and discussion section.
6- For readers to quickly catch your contribution, it would be better to highlight major difficulties and challenges, and your original achievements to overcome them, in a clearer way in abstract and introduction.
7- It is mentioned in “Main contribution of this research” that the authors employed DBN for feature extraction whereas NARX network was developed and trained for extrapolating the temporal forecasts. What are other feasible alternatives? What are the advantages of adopting this particular method over others in this case? How will this affect the results? The authors should provide more details on this.
8- More focus should be placed on the application of accurate multi-step rainfall forecasting. You can use specialized articles in this field.
* Emotional artificial neural networks (EANNs) for multi-step ahead prediction of monthly precipitation; case study: Northern Cyprus
* Threshold-based hybrid data mining method for long-term maximum precipitation forecasting
Accurate rainfall forecasting is very effective in predicting runoff, streamflow and flood:
* Emotional ANN (EANN) and Wavelet-ANN (WANN) Approaches for Markovian and Seasonal Based Modeling of Rainfall-Runoff Process
* Hybrid Wavelet-M5 Model Tree for Rainfall-Runoff Modeling
* Data mining based on wavelet and decision tree for rainfall-runoff simulation

9- Please explain that, how were the time series divided?
10- The titles to the figures and tables need to be much improved-more descriptive.
11- Please discuss the results and compare them with previous studies.
12- Please examine reference styles from "Instructions for Authors" in the journal website and arrange your references according to it.

Reviewer 2 ·

Basic reporting

The focus of this work is direct prediction of multistep forecasting, where a separate time series model for each forecasting horizon is considered and forecasts are computed using
observed data samples.

Experimental design

Forecasting in this method is performed by proposing a deep learning approach, i.e, Temporal Deep Belief Network (DBN). The best model is selected from several baseline models on the basis of performance analysis metrics. The results suggest that the temporal DBN model outperforms the conventional Convolutional Neural Network (CNN) specifically on rainfall time series forecasting.

Validity of the findings

The results are valid.

Additional comments

These are my main concerns:
1: The abstract is poor and it should be rewritten completely. Remove the unnecessary information from the abstract. Stay focus more on your work and contribution.
2: What is meant by one-step, four step and eight step? Explain this clearly whether this means by hourly, daily, weekly or monthly time step.
3: The introduction and background literature is not enough on rainfall using ML models. The authors should clearly work on it to enrich the literature section. For example the authors can cite the following relevant work.
a): Complete ensemble empirical mode decomposition hybridized with random forest and kernel ridge regression model for monthly rainfall forecasts; Journal of Hydrology 584, 124647, 2020.
b): Multi-stage hybridized online sequential extreme learning machine integrated with Markov Chain Monte Carlo copula-Bat algorithm for rainfall forecasting, Atmospheric research 213, 450-464, 2018.
c): Forecasting long-term precipitation for water resource management: a new multi-step data-intelligent modelling approach, Hydrological Sciences Journal, Volume 65 Issue 16 (2020).
d): An ensemble-ANFIS based uncertainty assessment model for forecasting multi-scalar standardized precipitation index, Atmospheric Research 207, 155-180, 2018.
e): Improving SPI-derived drought forecasts incorporating synoptic-scale climate indices in multi-phase multivariate empirical mode decomposition model hybridized with simulated. Journal of Hydrology 576, Pages 164-184, 2019.
f): Multi-stage committee based extreme learning machine model incorporating the influence of climate parameters and seasonality on drought forecasting. Computers and Electronics in Agriculture, Volume 152, September 2018, Pages 149-16

4: The presentation of the paper should improved. Some figures are hardly readable.
5: The data should be explained clearly. What are the input attributes that are used to predict the rainfall? How many data points are used for model training and testing purposes?
6: The experimental setup and model architectures should be discussed in details. They are very important.
7: The results should be used to explain in some other graphical visualization such as Taylor diagram, scatter plots etc.

---

## Round 0.2 · accepted · Accept

As per the feedback of the reviewers, the authors have successfully addressed all the required comments.

Reviewer 1 ·

Basic reporting

The authors answered to all of my concerns and comments

Experimental design

The authors answered to all of my concerns and comments

Validity of the findings

The authors answered to all of my concerns and comments

Additional comments

The authors answered to all of my concerns and comments

Reviewer 2 ·

Basic reporting

The authors have successfully revised the manuscript according to my feedback which improves to the level of publishing. I recommend to accept the paper in present form.

Experimental design

The authors have successfully revised the manuscript according to my feedback which improves to the level of publishing. I recommend to accept the paper in present form.

Validity of the findings

The authors have successfully revised the manuscript according to my feedback which improves to the level of publishing. I recommend to accept the paper in present form.

Additional comments

The authors have successfully revised the manuscript according to my feedback which improves to the level of publishing. I recommend to accept the paper in present form.